

# Arthropods of the great indoors: characterizing diversity inside urban and suburban homes

Matthew A. Bertone[1], Misha Leong[2], Keith M. Bayless[1], Tara L.F. Malow[3], Robert R. Dunn[4,5] and Michelle D. Trautwein[2]

[1] Department of Entomology, North Carolina State University, Raleigh, NC, United States of America
[2] California Academy of Sciences, San Francisco, CA, United States of America
[3] North Carolina Museum of Natural Sciences, Raleigh, NC, United States of America
[4] Department of Applied Ecology, North Carolina State University, Raleigh, NC, United States of America
[5] Center for Macroecology, Evolution and Climate, Natural History Museum of Denmark, University of Copenhagen, Copenhagen, Denmark

Corresponding author
Matthew A. Bertone,
matt_bertone@ncsu.edu,
matthew.bertone@gmail.com

## ABSTRACT

Although humans and arthropods have been living and evolving together for all of our history, we know very little about the arthropods we share our homes with apart from major pest groups. Here we surveyed, for the first time, the complete arthropod fauna of the indoor biome in 50 houses (located in and around Raleigh, North Carolina, USA). We discovered high diversity, with a conservative estimate range of 32–211 morphospecies, and 24–128 distinct arthropod families per house. The majority of this indoor diversity (73%) was made up of true flies (Diptera), spiders (Araneae), beetles (Coleoptera), and wasps and kin (Hymenoptera, especially ants: Formicidae). Much of the arthropod diversity within houses did not consist of synanthropic species, but instead included arthropods that were filtered from the surrounding landscape. As such, common pest species were found less frequently than benign species. Some of the most frequently found arthropods in houses, such as gall midges (Cecidomyiidae) and book lice (Liposcelididae), are unfamiliar to the general public despite their ubiquity. These findings present a new understanding of the diversity, prevalence, and distribution of the arthropods in our daily lives. Considering their impact as household pests, disease vectors, generators of allergens, and facilitators of the indoor microbiome, advancing our knowledge of the ecology and evolution of arthropods in homes has major economic and human health implications.

## INTRODUCTION

For as long as humans have lived in fixed habitations there have been other organisms that dwell alongside us. We share our living spaces with a variety of invited and uninvited guests spanning the tree of life, from large vertebrates (e.g., pets and livestock) to microorganisms (*Martin et al., 2015*). The most diverse and abundant group of multicellular life found in homes, as well as on Earth more generally, is represented by arthropods.

Insects, spiders and their relatives have been living and evolving with humans for all of our history. It has been proposed that many arthropod species that are now associated with human houses were originally cave dwellers (e.g., bed bugs: Cimicidae) (*Balvín et al., 2012*). Evidence of arthropod vectors in caves inhabited by prehistoric people ca. 26,000 years ago suggests that pestiferous arthropods, such as blood-feeding kissing bugs (Reduviidae: Triatominae), lived alongside our ancestors (*Araújo et al., 2009*). Among the first examples of cave art is a depiction of a camel cricket (Rhaphidophoridae) (*Chopard, 1928*; *Belles, 1997*).

As human society changed over time, arthropods successfully—and rapidly—made use of our bodies and resources for food and shelter. Constructed houses, animal domestication, agriculture and the ability to store food (such as grains) brought different arthropod species into the domiciles and daily lives of humans. Arthropods are common fauna in domestic archaeological sites from Egypt (dating as far back at 1353 B.C.E), Israel, and Europe (Switzerland, Greenland, and the UK) (*Nielsen, Mahler & Rasmussen, 2000*; *Panagiotakopulu, 2001*; *Panagiotakopulu, 2003*; *Kislev, Hartmann & Galili, 2004*; *Panagiotakopulu, 2004*; *Kenward & Carrott, 2006*), with some species characteristic of stored food products and livestock, and others representative of local fauna.

In contrast to the simple dwellings early humans used, modern western houses are perceived of as being an environment largely devoid of animal life. However, arthropods thrive in homes, as evidenced by today's multi-billion dollar pest control industry. Houses today host many of the same pest groups found in archeological sites (see 'Results'), yet today's arthropod communities also reflect aspects of society's modernization. For example, with the advent of indoor plumbing, dung beetles (Scarabaeidae) are less prevalent indoors (as shown in this study), but drain-dwelling moth flies (Psychodidae) are likely more so. Also, as human society became more globalized through travel and trade, arthropod species that are closely associated with humans and their homes, such as the house fly (*Musca domestica* L.; *Legner & McCoy, 1966*), German cockroach (*Blattella germanica* L.), and fruit fly (or vinegar fly, *Drosophila melanogaster* Meigen; *Keller, 2007*) obtained worldwide distributions and, in some cases, even lack wild populations (e.g., the German cockroach; *Roth, 1985*; *Booth et al., 2010*). The influence of human society and our changing domiciles on the evolution of specific arthropod lineages is evident.

Research on indoor arthropod communities has focused almost exclusively on pests, with a particular emphasis on those of medical and economic importance such as cockroaches, termites, bed bugs, fleas, and mosquitoes (*Committee on Urban Pest Management et al., 1980*; *Robinson, 2005*). Unlike the species that threaten or bother us, very little is known about the myriad other arthropod species, many of them inconspicuous (and even unnoticed), that live with humans. The true interactions between these other species and humans—be it beneficial, neutral, or negative—remain largely unknown, as does their prevalence and distribution. In fact, a comprehensive survey of arthropod life in contemporary human houses has never before been carried out.

A systematic sampling of the complete arthropod fauna in the interior of homes and an understanding of the role that interior microhabitats play in determining the assemblage of arthropod communities are the first steps toward revealing ecological dynamics in a

vastly understudied system (*Baz & Monserrat, 1999*; *Dunn et al., 2013*). What are the most prevalent arthropod groups found in houses and how common are they among homes and rooms? Here we explored the composition of overall arthropod diversity, including both pest and non-pest species, in human dwellings. Through surveying 50 free-standing houses located in the North Carolina Piedmont region, we identified and characterized the overall diversity of arthropods found within these homes.

## MATERIALS & METHODS

We solicited volunteers owning or renting free-standing homes in Raleigh and neighboring areas of North Carolina, USA. The study area is located in the Piedmont of the state and characterized by red clay soils and deciduous/pine forests (with meadows and aquatic/semi-aquatic systems interspersed) among various urban and suburban development (Fig. 1). We randomly selected 50 homes/volunteers to visit from among participants who filled out an online questionnaire about the characteristics of their household and behavior of its residents. All homes included in the study were within a 30 mile radius of Raleigh's center (35.7719°N, 78.6389°W). Each home was visited once between May and October 2012. Upon arrival, volunteers were informed of the procedures and process for sampling arthropods and asked to sign a consent form (Supplemental Information 1).

### General arthropod sampling

We (trained entomologists) performed a visual inspection of each room and collected specimens by hand using forceps, aspirators and entomological nets. Only visible surfaces, including those accessible under and behind furniture, around baseboards, ceilings, and on shelves and other surfaces were sampled. We collected all arthropods or putative arthropods, including those from spider webs; both living and dead arthropods were collected in this manner into vials containing 95% ethanol. We did not collect all specimens of a given arthropod species. We designated each distinct room or area and labeled its vial with the room name (as best identified), floor type, and number of windows and doors to the outside of the house; doors between rooms were not quantified. We identified floor types as wood (including laminates, hardwoods, and other wood-like surfaces), linoleum (tiles or otherwise), tile (stone, concrete, or otherwise; not including linoleum) or carpet. The presence of small or large rugs on other surfaces was noted as well. Typical room categories included kitchens, bedrooms (sometimes specified as offices/dens because of their lack of running water, but not receiving the same amount of traffic as common areas), bathrooms, laundry/utility rooms (denoted when a washer and dryer were present), and common areas. Common areas consisted of large open areas that were not easily categorized, usually including dining rooms, living rooms, front rooms, hallways, etc. Closets were sampled and included with the room in which they opened. When a room was present on a floor other than the first/main floor, house level was recorded (e.g., "2nd Floor Bathroom"). All rooms inside the house were sampled in the manner described above except for attics and crawl spaces, which were sampled less thoroughly: only the entrance of each was sampled within a 2 m radius. The limited

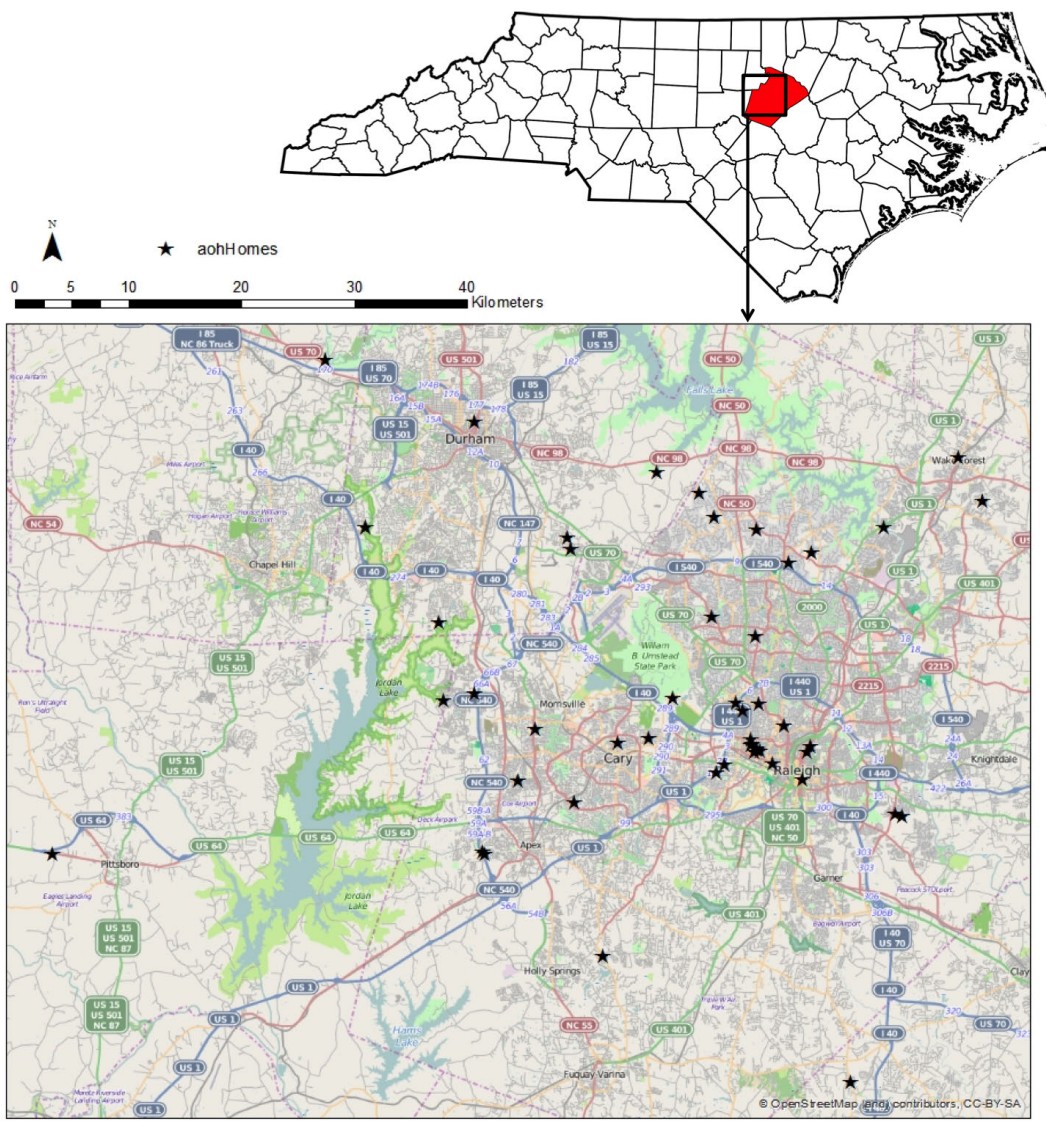

**Figure 1** **Map of study sites.** Fifty homes (denoted by stars) were selected in and around the Raleigh, North Carolina area. Raleigh is within Wake County, highlighted in red on the state map below, though some samples fell within adjacent counties.

sampling method for these areas was deemed necessary for the safety of those collecting the specimens (i.e., to avoid high summer temperatures in attics and confined areas in crawl spaces). Screened porches, decks, garages, detached sheds/structures, and closets accessible only from the outside were not sampled.

## Dust mites

We sampled dust mites from the middle of the master bedroom floor, regardless of floor type, in a 0.5 m² area. To collect mites, we used a vacuum that was adapted to use specimen cups modified with a screen bottom composed of mesh with 0.0055 inch (0.1397 mm) openings, small enough to allow air to pass through but not mites. We

stored all samples in 95% ethanol until they were sorted and quantified. Because of time and specimen handling constraints, only five individuals from five randomly-selected samples were identified.

## Identification and classification

We identified all specimens to family level except when specimens were badly damaged or required additional methods for identification (e.g., slide mounting of mites and other taxa). We further determined genera and species when possible, but many specimens could not be identified to such a level for several reasons including, but not limited to, being damaged, being an unidentifiable sex (e.g., female Sciaridae) or an unidentifiable life stage (e.g., larvae), or being a group that is understudied or lacking good diagnostic keys. As such, our approach produced a very conservative list of morphotaxa for each room in the house and hereafter we call these morphospecies; this type of characterization has been found to be effective in comparing species richness and turnover between sites (*Oliver & Beattie, 2002*). The taxonomic identity of morphospecies was not compared between rooms within homes due to limitations on time, space (storage of voucher specimens) and diagnostic expertise, thus the true number of morphospecies per house and among all houses was not definitively determined. Thus, a conservative (or assumed minimum) estimate of morphospecies richness per house was created by taking the maximum number of morphospecies from the room containing the highest number of morphospecies, for each family, and summing the total. This is in contrast to a maximum estimate of morphospecies richness which was the total sum of all morphospecies from all rooms within a house; this maximum corresponds to the case in which no two rooms held the same morphospecies. We assume the true diversity falls between our minimum and maximum estimates. All voucher specimens are housed in vials of ethanol in the laboratory of RRD (Department of Biological Sciences, NCSU) for use in further ecological, genetic, and microbiological studies. Specimens will be deposited in the insect museum at NCSU (Department of Entomology) when permanently housed.

## Analyses

We classified rooms into 6 categories based on their similarities of features and use: attics, basements (including finished and unfinished basements, and crawl spaces), bathrooms (including bathrooms and laundry rooms), bedrooms (including bedrooms, offices, and libraries), common rooms (including living rooms, dining rooms, and attached hallways), and kitchens (including kitchens and pantries); rooms not conforming to one of the categories were classified as ''other'' and were excluded from Table 1. We estimated diversity of families based on the complete list of families acquired over each sampled house using the Chao2 Estimator with 1,000 randomization runs in EstimateS (*Colwell, 2013*). We compared total dust mite abundance in each sample with the floor type on which it was collected. Some samples were collected on carpet, while others were on bare surfaces. Because the data were not normally distributed, we analyzed the differences between samples with different floor types using a Kruskal–Wallis test. All analyses were done in R 3.1.2 (*R Core Team, 2015*).

Bertone et al. (2016), *PeerJ*, DOI 10.7717/peerj.1582

**Table 1  List of arthropods found during the study present in at least 10% (*n* = 5) of homes.** Table includes the percentage of homes, rooms overall, and six specific room types (attics, basements, bathrooms, bedrooms, common areas, and kitchens) where a taxon was collected. One hundred twenty-eight additional families were collected, but were found in less than 10% (*n* = 5) of homes. For the full table containing all taxa (including genera and species that were identified) see Table S1. All names are based on current taxonomy except for mites, where "Acari" is used as a general order despite modern classifications that consider the group a subclass with numerous orders (*Krantz & Walter, 2009*). s.l. = *sensu lato*, i.e., "broad sense."

| Class | Order | Family | Common name | Homes (*n* = 50) | Rooms (*n* = 554) | Attic (*n* = 38) | Basement (*n* = 54) | Bath (*n* = 146) | Bed (*n* = 160) | Common (*n* = 97) | Kitchen (*n* = 50) |
|---|---|---|---|---|---|---|---|---|---|---|---|
| **Entognatha (non-insect hexapods)** | | | | **88%** | **22.9%** | **5.3%** | **25.9%** | **15.8%** | **21.9%** | **37.1%** | **26.0%** |
| | **Collembola (springtails)** | | | **88%** | **22.9%** | **5.3%** | **25.9%** | **15.8%** | **21.9%** | **37.1%** | **26.0%** |
| | | Entomobryidae | *slender springtails* | 78% | 19.1% | 5.3% | 20.4% | 15.1% | 16.9% | 28.9% | 24.0% |
| | | Tomoceridae | *elongate springtails* | 26% | 3.6% | – | 1.9% | 0.7% | 3.8% | 9.3% | 4.0% |
| **Insecta (true insects)** | | | | **100%** | **94.9%** | **78.9%** | **96.3%** | **91.1%** | **98.1%** | **97.9%** | **100.0%** |
| | **Archaeognatha (jumping bristletails)** | | | **18%** | **2.5%** | **–** | **1.9%** | **2.7%** | **1.9%** | **5.2%** | **2.0%** |
| | | Machilidae | *jumping bristletails* | 14% | 2.0% | – | 1.9% | 1.4% | 1.3% | 5.2% | 2.0% |
| | **Zygentoma (silverfish)** | | | **68%** | **21.3%** | **21.1%** | **3.7%** | **15.1%** | **23.1%** | **40.2%** | **16.0%** |
| | | Lepismatidae | *silverfish & firebrats* | 68% | 21.3% | 21.1% | 3.7% | 15.1% | 23.1% | 40.2% | 16.0% |
| | **Dermaptera (earwigs)** | | | **50%** | **7.6%** | **–** | **16.7%** | **4.1%** | **6.3%** | **14.4%** | **6.0%** |
| | | Anisolabididae | *earwigs* | 26% | 3.6% | – | 9.3% | 2.7% | 2.5% | 6.2% | 2.0% |
| | | Forficulidae | *earwigs* | 24% | 3.4% | – | 7.4% | 1.4% | 3.1% | 6.2% | 4.0% |
| | **Orthoptera (grasshoppers, crickets & katydids)** | | | **76%** | **17.7%** | **2.6%** | **53.7%** | **7.5%** | **11.3%** | **29.9%** | **14.0%** |
| | | Gryllidae | *crickets* | 30% | 3.6% | – | 5.6% | 2.1% | 3.8% | 8.2% | – |
| | | Myrmecophilidae | *ant-loving crickets* | 10% | 0.9% | – | – | – | – | 2.1% | 6.0% |
| | | Rhaphidophoridae | *camel & cave crickets* | 58% | 12.8% | – | 50.0% | 5.5% | 6.9% | 19.6% | 6.0% |
| | **Blattodea (cockroaches)** | | | **82%** | **25.5%** | **18.4%** | **33.3%** | **14.4%** | **26.3%** | **43.3%** | **18.0%** |
| | | Blattidae | *cockroaches* | 74% | 22.9% | 15.8% | 33.3% | 12.3% | 22.5% | 40.2% | 16.0% |
| | | Ectobiidae | *cockroaches* | 34% | 4.3% | 2.6% | 5.6% | 2.1% | 3.8% | 10.3% | 2.0% |
| | **Isoptera (termites)** | | | **28%** | **4.9%** | **–** | **3.7%** | **1.4%** | **4.4%** | **12.4%** | **6.0%** |
| | | Rhinotermitidae | *subterranean termites* | 28% | 4.9% | – | 3.7% | 1.4% | 4.4% | 12.4% | 6.0% |
| | **Hemiptera (true bugs)** | | | **98%** | **36.3%** | **7.9%** | **27.8%** | **18.5%** | **36.9%** | **68.0%** | **58.0%** |
| | | Anthocoridae | *minute pirate bugs* | 42% | 6.0% | – | – | 2.7% | 5.6% | 16.5% | 8.0% |
| | | Aphididae | *aphids* | 56% | 7.6% | 2.6% | 7.4% | 1.4% | 6.3% | 15.5% | 20.0% |
| | | Cicadellidae | *leafhoppers* | 82% | 16.2% | – | 5.6% | 8.9% | 13.8% | 37.1% | 30.0% |
| | | Coreidae | *leaf-footed bugs* | 12% | 1.1% | 2.6% | – | – | 1.3% | 3.1% | – |
| | | Cydnidae | *burrowing bugs* | 28% | 2.9% | 2.6% | 3.7% | 1.4% | 1.9% | 8.2% | – |
| | | Delphacidae | *delphacid planthoppers* | 12% | 1.4% | – | 1.9% | 1.4% | 0.6% | 3.1% | – |
| | | Lygaeidae | *seed bugs* | 10% | 1.4% | – | – | 0.7% | 1.3% | 4.1% | – |
| | | Miridae | *plant bugs* | 44% | 6.5% | – | – | 2.1% | 5.6% | 19.6% | 8.0% |

Bertone et al. (2016), *PeerJ*, DOI 10.7717/peerj.1582

**Table 1** (*continued*)

| Class | Order | Family | Common name | Homes (*n* = 50) | Rooms (*n* = 554) | Attic (*n* = 38) | Basement (*n* = 54) | Bath (*n* = 146) | Bed (*n* = 160) | Common (*n* = 97) | Kitchen (*n* = 50) |
|---|---|---|---|---|---|---|---|---|---|---|---|
| | | Pentatomidae | *stink bugs* | 22% | 2.5% | 5.3% | – | – | 3.1% | 5.2% | 4.0% |
| | | Psyllidae s.l. | *jumping plant lice* | 10% | 0.9% | – | – | 0.7% | – | 3.1% | 2.0% |
| | | Reduviidae | *assassin bugs* | 28% | 3.4% | – | 3.7% | – | 1.9% | 11.3% | 4.0% |
| | | Rhyparochromidae | *dirt-colored seed bugs* | 30% | 3.8% | – | 3.7% | 4.1% | 1.9% | 9.3% | 2.0% |
| | | Tingidae | *lace bugs* | 16% | 1.6% | – | 1.9% | 1.4% | 1.3% | 4.1% | – |
| | **Psocodea (lice)** | | | **98%** | **43.1%** | **36.8%** | **24.1%** | **31.5%** | **50.6%** | **60.8%** | **46.0%** |
| | | Ectopsocidae | *bark lice* | 16% | 2.0% | – | – | – | 2.5% | 5.2% | 4.0% |
| | | Lepidopsocidae | *scaly-winged bark lice* | 24% | 4.3% | 2.6% | – | 0.7% | 5.6% | 13.4% | – |
| | | Liposcelididae | *book lice* | 98% | 37.4% | 34.2% | 16.7% | 28.1% | 44.4% | 52.6% | 38.0% |
| | **Thysanoptera (thrips)** | | | **50%** | **7.0%** | **2.6%** | **5.6%** | **2.7%** | **4.4%** | **19.6%** | **8.0%** |
| | | Phlaeothripidae | *tube-tailed thrips* | 14% | 1.4% | – | 1.9% | 0.7% | 0.6% | 5.2% | – |
| | | Thripidae | *common thrips* | 32% | 4.2% | 2.6% | – | 2.1% | 2.5% | 11.3% | 6.0% |
| | **Hymenoptera (wasps, ants & bees)** | | | **100%** | **69.7%** | **50.0%** | **64.8%** | **50.7%** | **78.8%** | **85.6%** | **84.0%** |
| | | Bethylidae | *bethylid wasps* | 28% | 3.8% | – | 1.9% | 0.7% | 2.5% | 14.4% | 2.0% |
| | | Braconidae | *braconid wasps* | 52% | 7.6% | – | – | 4.1% | 8.8% | 19.6% | 6.0% |
| | | Ceraphronidae | *ceraphronid wasps* | 14% | 1.4% | – | 1.9% | – | – | 7.2% | – |
| | | Chalcididae | *chalcidid wasps* | 14% | 1.4% | – | – | 0.7% | 1.9% | 4.1% | – |
| | | Diapriidae | *diapriid wasps* | 26% | 2.5% | – | 5.6% | – | 1.3% | 8.2% | 2.0% |
| | | Encyrtidae | *encyrtid wasps* | 12% | 1.4% | – | – | – | 1.9% | 4.1% | – |
| | | Eulophidae | *eulophid wasps* | 70% | 17.0% | – | 7.4% | 13.0% | 16.9% | 30.9% | 22.0% |
| | | Formicidae | *ants* | 100% | 61.9% | 47.4% | 57.4% | 41.8% | 66.3% | 81.4% | 82.0% |
| | | Halictidae | *sweat bees* | 10% | 0.9% | 2.6% | 1.9% | 0.7% | – | – | 4.0% |
| | | Ichneumonidae | *ichneumon wasps* | 38% | 4.5% | 2.6% | 5.6% | 1.4% | 4.4% | 9.3% | 4.0% |
| | | Mymaridae | *fairyflies* | 26% | 2.3% | – | – | 0.7% | – | 11.3% | 2.0% |
| | | Platygastridae s.l. | *platygastrid wasps* | 58% | 8.3% | – | – | 6.2% | 5.6% | 27.8% | 2.0% |
| | | Pompilidae | *spider wasps* | 34% | 6.5% | 2.6% | 7.4% | 2.7% | 5.6% | 17.5% | – |
| | | Pteromalidae | *pteromalid wasps* | 42% | 4.9% | – | – | 2.1% | 3.8% | 10.3% | 16.0% |
| | | Sphecidae s.l. | *thread-waisted wasps* | 26% | 3.2% | 2.6% | 1.9% | 2.7% | 3.8% | 6.2% | – |
| | | Vespidae | *paper wasps & hornets* | 14% | 1.6% | 5.3% | 1.9% | - | 1.9% | 3.1% | – |
| | **Neuroptera (lacewings, antlions, etc.)** | | | **56%** | **6.9%** | **–** | **3.7%** | **2.7%** | **4.4%** | **20.6%** | **10.0%** |
| | | Chrysopidae | *green lacewings* | 34% | 3.6% | – | 1.9% | 1.4% | 2.5% | 11.3% | 4.0% |
| | | Coniopterygidae | *dustywings* | 16% | 1.4% | – | – | 0.7% | 1.3% | 4.1% | 2.0% |
| | | Hemerobiidae | *brown lacewings* | 18% | 1.8% | – | – | – | – | 7.2% | 6.0% |
| | **Coleoptera (beetles)** | | | **100%** | **72.0%** | **44.7%** | **64.8%** | **54.1%** | **84.4%** | **91.8%** | **72.0%** |
| | | Aderidae | *ant-like leaf beetles* | 22% | 2.5% | – | – | 0.7% | 3.1% | 5.2% | 6.0% |
| | | Anobiidae | *death watch beetles* | 60% | 12.1% | – | 7.4% | 5.5% | 10.6% | 30.9% | 14.0% |

Peerj

| Class | Order | Family | Common name | Homes (n = 50) | Rooms (n = 554) | Attic (n = 38) | Basement (n = 54) | Bath (n = 146) | Bed (n = 160) | Common (n = 97) | Kitchen (n = 50) |
|---|---|---|---|---|---|---|---|---|---|---|---|
| | | Anthicidae | ant-like flower beetles | 18% | 2.3% | 2.6% | – | 0.7% | 1.9% | 7.2% | – |
| | | Carabidae | ground beetles | 66% | 9.9% | 7.9% | 33.3% | 4.1% | 6.9% | 14.4% | 2.0% |
| | | Cerambycidae | longhorned beetles | 16% | 1.6% | – | – | 0.7% | 3.1% | 3.1% | – |
| | | Chrysomelidae | leaf beetles | 46% | 6.0% | – | – | 1.4% | 8.1% | 17.5% | 2.0% |
| | | Cleridae | checkered beetles | 18% | 1.8% | – | – | 1.4% | 3.1% | 3.1% | – |
| | | Coccinellidae | ladybugs | 52% | 7.8% | 5.3% | 1.9% | 0.7% | 8.8% | 22.7% | 4.0% |
| | | Cryptophagidae | silken fungus beetles | 26% | 3.2% | – | – | – | 2.5% | 12.4% | 4.0% |
| | | Curculionidae | weevils, bark beetles | 82% | 15.7% | 2.6% | 18.5% | 9.6% | 16.3% | 34.0% | 6.0% |
| | | Dermestidae | carpet beetles | 100% | 57.0% | 26.3% | 22.2% | 44.5% | 71.3% | 82.5% | 58.0% |
| | | Elateridae | click beetles | 74% | 14.6% | 5.3% | 13.0% | 5.5% | 12.5% | 39.2% | 8.0% |
| | | Histeridae | clown beetles | 10% | 0.9% | – | 1.9% | 0.7% | 0.6% | 2.1% | – |
| | | Lampyridae | fireflies | 20% | 2.2% | 2.6% | 1.9% | – | 1.3% | 6.2% | 4.0% |
| | | Latridiidae | minute brown scavenger beetles | 38% | 6.3% | – | 1.9% | 1.4% | 5.6% | 16.5% | 12.0% |
| | | Melyridae | soft-winged flower beetles | 20% | 2.9% | – | 1.9% | 2.7% | 3.1% | 5.2% | 2.0% |
| | | Mordellidae | tumbling flower beetles | 24% | 3.6% | 2.6% | 3.7% | 0.7% | 3.1% | 8.2% | 4.0% |
| | | Mycetophagidae | hairy fungus beetles | 20% | 1.8% | 2.6% | – | – | 2.5% | 4.1% | 2.0% |
| | | Nitidulidae | sap beetles | 24% | 3.1% | – | 3.7% | – | 2.5% | 10.3% | 2.0% |
| | | Phalacridae | shining flower beetles | 12% | 1.3% | – | 1.9% | – | 0.6% | 5.2% | – |
| | | Ptilodactylidae | ptilodactylid beetles | 30% | 4.9% | 2.6% | 3.7% | 2.1% | 5.6% | 9.3% | 6.0% |
| | | Scarabaeidae | scarab beetles | 52% | 9.4% | 5.3% | 13.0% | 2.7% | 8.1% | 20.6% | 10.0% |
| | | Scraptiidae | false flower beetles | 20% | 2.0% | – | – | 0.7% | 1.9% | 6.2% | 2.0% |
| | | Silvanidae | flat bark beetles | 46% | 6.5% | – | 9.3% | 2.1% | 5.0% | 17.5% | 4.0% |
| | | Staphylinidae | rove beetles | 54% | 7.2% | – | 7.4% | 4.8% | 5.0% | 17.5% | 6.0% |
| | | Tenebrionidae | darkling beetles | 62% | 11.2% | 5.3% | 16.7% | 4.8% | 11.3% | 24.7% | 2.0% |
| | | Throscidae | false metallic wood boring beetles | 22% | 2.7% | – | 1.9% | 0.7% | 1.9% | 9.3% | 2.0% |
| | | Trogossitidae | bark gnawing beetles | 16% | 1.4% | – | – | 0.7% | 0.6% | 6.2% | – |
| | | Zopheridae | ironclad beetles | 16% | 1.4% | – | – | – | 2.5% | 1.0% | 6.0% |
| | **Lepidoptera (moths & butterflies)** | | | **92%** | **28.7%** | **–** | **16.7%** | **11.6%** | **33.8%** | **56.7%** | **38.0%** |
| | | Geometridae | geometrid moths | 12% | 1.6% | – | 1.9% | 0.7% | 1.9% | 2.1% | 2.0% |
| | | Noctuidae | owlet moths | 44% | 5.8% | – | 3.7% | 4.1% | 5.0% | 12.4% | 6.0% |
| | | Pyralidae | pyralid moths | 62% | 11.0% | – | 3.7% | 4.8% | 12.5% | 21.6% | 22.0% |
| | | Tineidae | clothes moths | 60% | 8.8% | – | – | 5.5% | 9.4% | 19.6% | 10.0% |
| | | Tortricidae | leafroller moths | 10% | 1.1% | – | – | 0.7% | 0.6% | 4.1% | – |
**Table 1** (*continued*)

| Class | Order | Family | Common name | Homes (*n* = 50) | Rooms (*n* = 554) | Attic (*n* = 38) | Basement (*n* = 54) | Bath (*n* = 146) | Bed (*n* = 160) | Common (*n* = 97) | Kitchen (*n* = 50) |
|---|---|---|---|---|---|---|---|---|---|---|---|
| | **Trichoptera (caddisflies)** | | | **12%** | **1.1%** | – | – | **0.7%** | **1.3%** | **3.1%** | – |
| | **Siphonaptera (fleas)** | | | **10%** | **1.4%** | – | – | **2.7%** | **1.3%** | **1.0%** | **2.0%** |
| | | Pulicidae | *cat, dog & human fleas* | 10% | 1.4% | – | – | 2.7% | 1.3% | 1.0% | 2.0% |
| | **Diptera (true flies)** | | | **100%** | **71.8%** | **18.4%** | **50.0%** | **66.4%** | **85.6%** | **85.6%** | **80.0%** |
| | | Agromyzidae | *leafminer flies* | 12% | 1.1% | – | – | 0.7% | 0.6% | 3.1% | 2.0% |
| | | Anisopodidae | *wood gnats* | 10% | 0.9% | – | – | – | 0.6% | 3.1% | 2.0% |
| | | Anthomyiidae | *root maggot flies* | 10% | 0.9% | – | – | – | 1.3% | 1.0% | 4.0% |
| | | Bibionidae | *march flies, lovebugs* | 26% | 2.7% | – | – | 0.7% | 3.8% | 7.2% | 2.0% |
| | | Calliphoridae | *blow flies* | 48% | 8.1% | – | 5.6% | 2.7% | 7.5% | 23.7% | 6.0% |
| | | Cecidomyiidae | *gall midges* | 100% | 36.1% | 10.5% | 16.7% | 30.1% | 33.8% | 63.9% | 46.0% |
| | | Ceratopogonidae | *biting midges* | 54% | 7.6% | – | – | 3.4% | 8.8% | 18.6% | 10.0% |
| | | Chaoboridae | *phantom midges* | 14% | 3.4% | – | – | 3.4% | 1.9% | 8.2% | 6.0% |
| | | Chironomidae | *non-biting midges* | 80% | 17.0% | – | 3.7% | 8.9% | 13.8% | 42.3% | 30.0% |
| | | Chloropidae | *frit flies* | 28% | 4.3% | – | – | 3.4% | 3.1% | 10.3% | 8.0% |
| | | Culicidae | *mosquitoes* | 82% | 19.0% | – | 5.6% | 7.5% | 24.4% | 41.2% | 22.0% |
| | | Dolichopodidae | *longlegged flies* | 44% | 4.9% | – | 3.7% | 2.7% | 4.4% | 12.4% | 4.0% |
| | | Drosophilidae | *fruit flies, vinegar flies* | 66% | 13.7% | – | 5.6% | 4.8% | 15.6% | 29.9% | 22.0% |
| | | Empididae s.l. | *dance flies* | 16% | 1.8% | – | 1.9% | 0.7% | 1.3% | 6.2% | – |
| | | Ephydridae | *shore flies* | 14% | 1.4% | – | – | 1.4% | – | 6.2% | – |
| | | Fanniidae | *lesser house flies* | 10% | 1.1% | – | 1.9% | 0.7% | 0.6% | 1.0% | 4.0% |
| | | Lauxaniidae | *lauxaniid flies* | 16% | 1.6% | – | – | 0.7% | 1.3% | 6.2% | – |
| | | Milichiidae | *freeloader flies* | 14% | 1.6% | – | – | 0.7% | 1.9% | 4.1% | – |
| | | Muscidae | *house & stable flies* | 44% | 6.3% | 2.6% | 5.6% | 1.4% | 5.6% | 16.5% | 4.0% |
| | | Mycetophilidae s.l. | *fungus gnats* | 68% | 16.2% | – | 9.3% | 8.2% | 20.6% | 37.1% | 6.0% |
| | | Phoridae | *scuttle flies* | 82% | 17.3% | – | 20.4% | 6.2% | 16.9% | 39.2% | 18.0% |
| | | Psychodidae | *moth flies* | 74% | 18.8% | – | 9.3% | 14.4% | 18.1% | 41.2% | 14.0% |
| | | Sarcophagidae | *flesh flies* | 38% | 5.1% | – | – | 2.7% | 4.4% | 12.4% | 6.0% |
| | | Scatopsidae | *minute black scavenger flies* | 50% | 6.9% | – | 1.9% | 5.5% | 4.4% | 16.5% | 10.0% |
| | | Sciaridae | *dark-winged fungus gnats* | 96% | 42.1% | – | 22.2% | 35.6% | 49.4% | 64.9% | 42.0% |
| | | Sphaeroceridae | *lesser dung flies* | 28% | 3.4% | 2.6% | 11.1% | 2.1% | 2.5% | 5.2% | – |
| | | Stratiomyidae | *soldier flies* | 22% | 2.3% | – | 1.9% | 0.7% | 0.6% | 8.2% | 2.0% |
| | | Tachinidae | *tachinid flies* | 18% | 1.6% | – | – | – | 1.9% | 5.2% | 2.0% |
| | | Tipulidae s.l. | *crane flies* | 74% | 15.9% | 2.6% | 7.4% | 11.0% | 16.9% | 32.0% | 18.0% |
| | | Trichoceridae | *winter crane flies* | 20% | 2.3% | – | 1.9% | 2.1% | 1.9% | 6.2% | – |

Bertone et al. (2016), *PeerJ*, DOI 10.7717/peerj.1582

**Table 1** (*continued*)

| Class | Order | Family | Common name | Homes (*n* = 50) | Rooms (*n* = 554) | Attic (*n* = 38) | Basement (*n* = 54) | Bath (*n* = 146) | Bed (*n* = 160) | Common (*n* = 97) | Kitchen (*n* = 50) |
|---|---|---|---|---|---|---|---|---|---|---|---|
| **Arachnida (arachnids)** | | | | **100%** | **79.6%** | **47.4%** | **94.4%** | **68.5%** | **78.1%** | **96.9%** | **90.0%** |
| | **Araneae (spiders)** | | | **100%** | **78.5%** | **47.4%** | **92.6%** | **68.5%** | **75.6%** | **96.9%** | **88.0%** |
| | | Agelenidae | *funnel weavers, grass spiders* | 46% | 8.3% | 2.6% | 20.4% | 2.1% | 6.9% | 16.5% | 6.0% |
| | | Anyphaenidae | *ghost spiders* | 30% | 4.3% | – | 1.9% | 0.7% | 5.6% | 11.3% | 4.0% |
| | | Araneidae | *orb weavers* | 18% | 2.2% | – | 1.9% | 1.4% | 1.3% | 6.2% | 2.0% |
| | | Clubionidae | *sac spiders* | 10% | 0.9% | – | – | – | – | 4.1% | 2.0% |
| | | Corinnidae | *antmimics, ground spiders* | 38% | 6.0% | 5.3% | 9.3% | 1.4% | 6.9% | 11.3% | 4.0% |
| | | Gnaphosidae | *ground spiders* | 48% | 8.1% | – | 5.6% | 3.4% | 10.0% | 20.6% | 2.0% |
| | | Linyphiidae | *sheetweb & dwarf spiders* | 22% | 2.9% | – | 3.7% | – | 1.3% | 10.3% | 2.0% |
| | | Lycosidae | *wolf spiders* | 40% | 5.8% | 2.6% | 5.6% | 2.1% | 2.5% | 16.5% | 8.0% |
| | | Oecobiidae | *wall spiders* | 28% | 8.8% | 2.6% | – | 6.2% | 10.0% | 18.6% | 8.0% |
| | | Oonopidae | *goblin spiders* | 16% | 3.2% | 5.3% | – | 2.1% | 3.8% | 6.2% | 2.0% |
| | | Pholcidae | *cellar spiders* | 84% | 28.0% | 7.9% | 38.9% | 19.2% | 18.1% | 56.7% | 32.0% |
| | | Salticidae | *jumping spiders* | 50% | 8.3% | 2.6% | 3.7% | 2.7% | 7.5% | 22.7% | 6.0% |
| | | Scytodidae | *spitting spiders* | 16% | 2.7% | – | 1.9% | 3.4% | 2.5% | 4.1% | 2.0% |
| | | Theridiidae | *cobweb spiders* | 100% | 65.3% | 39.5% | 77.8% | 55.5% | 61.3% | 87.6% | 70.0% |
| | | Thomisidae | *crab spiders* | 32% | 3.4% | 2.6% | 1.9% | – | 4.4% | 9.3% | 2.0% |
| | **"Acari" (mites)** | | | **76%** | **18.6%** | **5.3%** | **46.3%** | **4.8%** | **17.5%** | **36.1%** | **10.0%** |
| | | Galumnidae | *armored mites* | 12% | 1.1% | – | 1.9% | – | 0.6% | 4.1% | – |
| | | Ixodidae | *hard ticks* | 18% | 2.0% | – | – | 0.7% | 1.9% | 7.2% | – |
| | | UnID Oribatida | *armored mites* | 46% | 6.0% | – | 22.2% | 1.4% | 3.8% | 12.4% | 2.0% |
| | | Pyroglyphidae | *dust mites* | 76% | NA | NA | NA | NA | NA | NA | NA |
| | **Opiliones (harvestmen & daddy-longlegs)** | | | **16%** | **2.3%** | **2.6%** | **3.7%** | **0.7%** | **2.5%** | **5.2%** | **-** |
| | **Pseudoscorpionida (pseudoscorpions)** | | | **20%** | **2.7%** | **–** | **3.7%** | **0.7%** | **0.6%** | **10.3%** | **2.0%** |
| **Chilopoda (centipedes)** | | | | **42%** | **9.2%** | **–** | **16.7%** | **4.1%** | **9.4%** | **16.5%** | **10.0%** |
| | **Lithobiomorpha (stone centipedes)** | | | **18%** | **1.8%** | **–** | **1.9%** | **–** | **1.9%** | **5.2%** | **2.0%** |
| | | Lithobiidae | *stone centipedes* | 14% | 1.3% | – | 1.9% | – | 1.9% | 2.1% | 2.0% |
| | **Scolopendromorpha (tropical centipedes)** | | | **12%** | **2.2%** | **–** | **5.6%** | **0.7%** | **1.9%** | **5.2%** | **–** |
| | | Scolopendridae | *tropical centipedes* | 12% | 2.2% | – | 5.6% | 0.7% | 1.9% | 5.2% | – |
| | **Scutigeromorpha (house centipedes)** | | | **32%** | **6.9%** | **–** | **13.0%** | **3.4%** | **6.9%** | **10.3%** | **10.0%** |
| | | Scutigeridae | *house centipedes* | 32% | 6.9% | – | 13.0% | 3.4% | 6.9% | 10.3% | 10.0% |
| **Diplopoda (millipedes)** | | | | **82%** | **21.1%** | **5.3%** | **63.0%** | **6.2%** | **15.0%** | **38.1%** | **16.0%** |
| | **Callipodida (crested millipedes)** | | | **10%** | **1.8%** | **–** | **5.6%** | **–** | **2.5%** | **3.1%** | **–** |
| | | Abacionidae | *crested millipedes* | 10% | 1.8% | – | 5.6% | – | 2.5% | 3.1% | – |

Bertone et al. (2016), *PeerJ*, DOI 10.7717/peerj.1582

Peerj

**Table 1** (*continued*)

| Class | Order | Family | Common name | Homes (*n* = 50) | Rooms (*n* = 554) | Attic (*n* = 38) | Basement (*n* = 54) | Bath (*n* = 146) | Bed (*n* = 160) | Common (*n* = 97) | Kitchen (*n* = 50) |
|---|---|---|---|---|---|---|---|---|---|---|---|
| | **Julida (julid millipedes)** | | | **42%** | **4.7%** | **–** | **18.5%** | **2.1%** | **2.5%** | **9.3%** | **–** |
| | | Julidae | *millipedes* | 38% | 3.8% | – | 11.1% | 2.1% | 2.5% | 8.2% | – |
| | **Polydesmida (flat-backed millipedes)** | | | **72%** | **17.7%** | **5.3%** | **57.4%** | **4.8%** | **13.1%** | **29.9%** | **14.0%** |
| | | Paradoxosomatidae | *greenhouse millipedes* | 58% | 12.6% | 2.6% | 42.6% | 2.7% | 9.4% | 22.7% | 10.0% |
| | | Polydesmidae | *flat-backed millipedes* | 26% | 4.3% | – | 16.7% | 1.4% | 1.3% | 8.2% | 4.0% |
| | **Spirobolida (round-backed millipedes)** | | | **20%** | **2.3%** | **–** | **14.8%** | **0.7%** | **–** | **3.1%** | **2.0%** |
| | | Spirobolidae | *round-backed millipedes* | 18% | 2.2% | – | 14.8% | 0.7% | – | 2.1% | 2.0% |
| **Malacostraca (crustaceans)** | | | | **86%** | **23.8%** | **5.3%** | **57.4%** | **14.4%** | **20.0%** | **34.0%** | **22.0%** |
| | **Isopoda (isopods)** | | | **84%** | **23.6%** | **5.3%** | **57.4%** | **14.4%** | **20.0%** | **34.0%** | **20.0%** |
| | | Armadillidiidae | *pillbugs & roly polies* | 78% | 22.0% | 5.3% | 48.1% | 13.7% | 18.8% | 33.0% | 20.0% |
| | | Porcellionidae | *woodlice & sowbugs* | 20% | 4.2% | – | 14.8% | 1.4% | 3.8% | 5.2% | 4.0% |

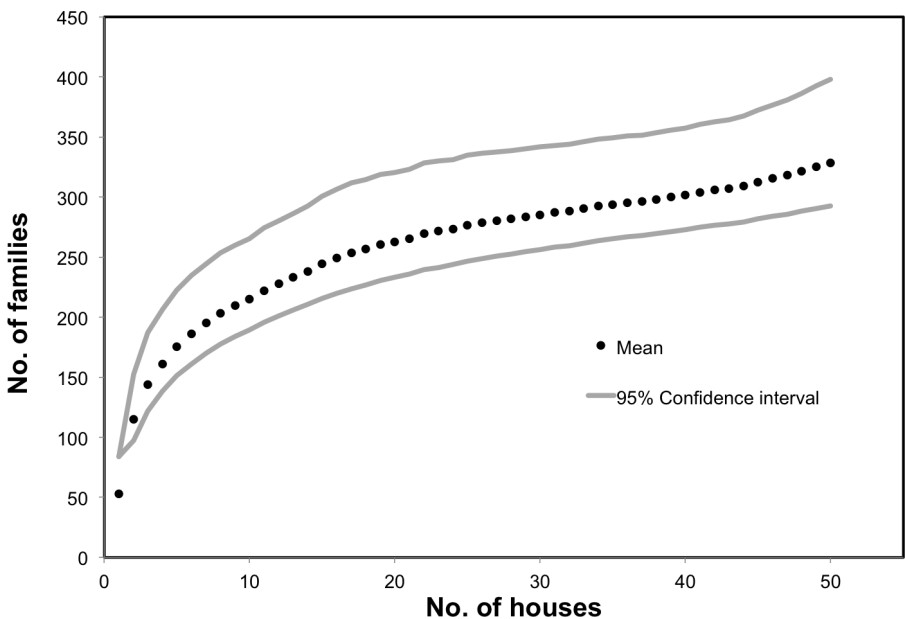

**Figure 2** **Estimated diversity of families.** The mean, along with 95% lower and upper confidence intervals, was calculated based on the complete list of families acquired over each sampled house using the Chao2 Estimator with 1,000 randomization runs in EstimateS (*Colwell, 2013*).

## RESULTS

### Overall metrics

Houses in the study ranged from 840 to 4,833 square feet in area (mean = 2,072; median = 1,720) and were from seven to 94 years old (mean = 41.35; median = 30.5). During the course of sampling 554 rooms in the 50 homes, over 10,000 specimens were collected and identified. These specimens represented all four subphyla (Chelicerata, Myriapoda, "Crustacea," and Hexapoda), as well as six classes, 34 orders and 304 families of arthropods (Table 1 and Table S1). While we cannot determine the exact number of morphospecies that were collected, there were at least 579 morphospecies based on our most conservative estimates (calculated by summing the maximum number of morphospecies for each family ever found in a single room).

We collected 24–128 families from each house, resulting in an average of 61.84 (s.d. = 23.24) distinct arthropod families per house and a total gamma diversity (across houses) of 304 families (Fig. 2). One hundred and forty-nine (149) families were rare, collected from fewer than 10% of homes, 66 of which were found in just a single home. The number of families collected in a home was correlated with house size ($r^2 = 0.3$, $p =< 0.001$). Conservative species estimates by home ranged from 32 to 211, with an average of 93.14 (s.d. = 42.34) morphospecies per house (Fig. 3). Considering that our conservative species estimate assumes that rooms with the greatest number of morphospecies by family included all species from other rooms (which is almost certainly untrue), this number is likely much lower than the true number of species per house (Fig. 3).

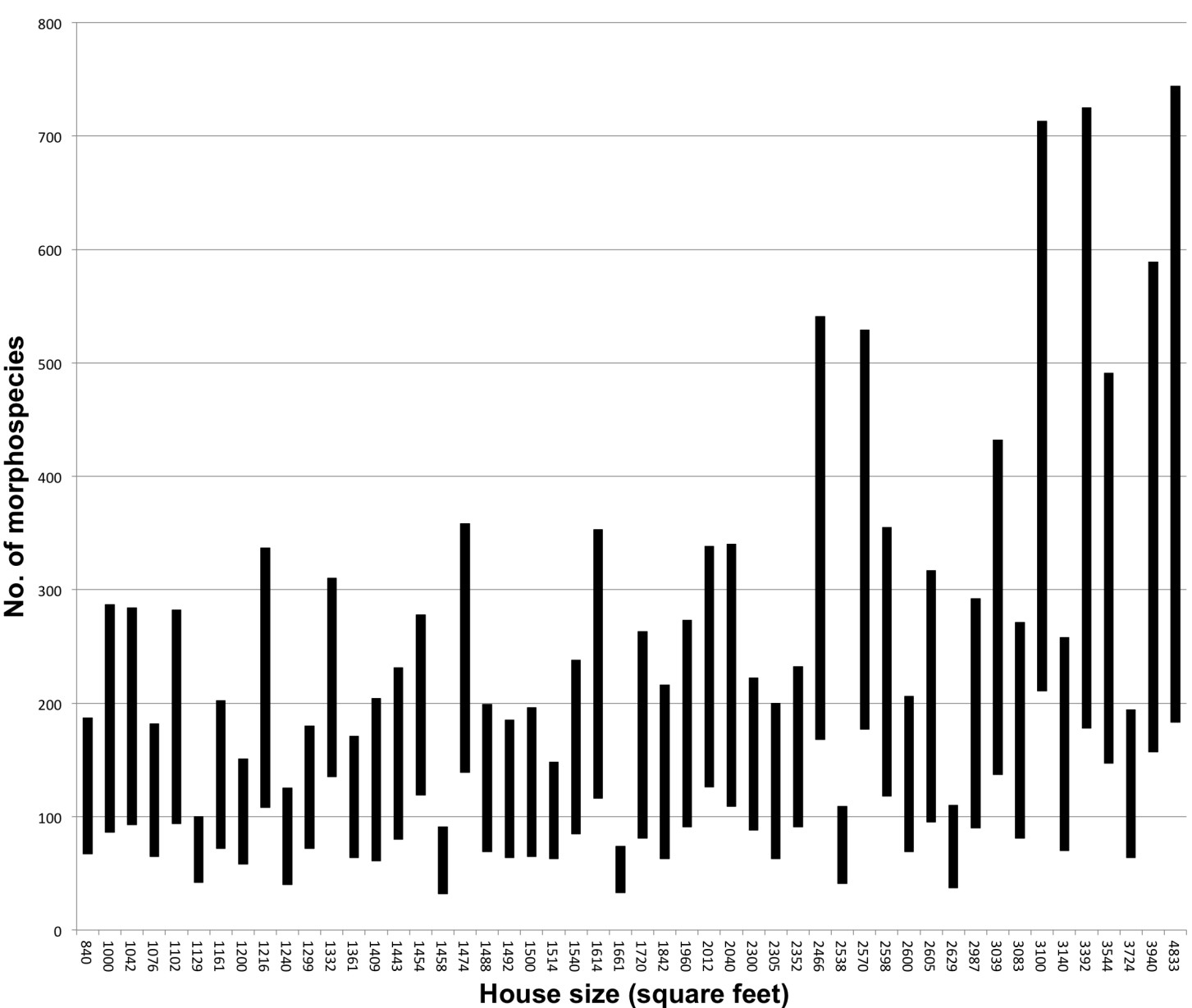

**Figure 3** **Number of species by house (in ascending rank order of house size).** The number of species collected by house is represented by the middle bar. The bottom limit is the minimum/conservative estimate of morphospecies by house which was calculated by taking the maximum number of morphospecies from the room containing the highest number of morphospecies, for each family, and summing the total The upper limit is the maximum possible of morphospecies within a house, with the assumption that each set of morphospecies within each room were unique from other rooms. Houses furthest to the left are the smallest in terms of square footage, and those furthest to the right are the largest. Houses ranged in size from 840 to 4,833 square feet.

## Taxon specific observations

While overall diversity was high, 12 frequently found families were identified in at least 80% of homes (Fig. 4). Only four families were identified from 100% of houses sampled: cobweb spiders (Theridiidae), carpet beetles (Dermestidae), gall midge flies (Cecidomyiidae) and ants (Formicidae). Book lice (Liposcelididae) and dark-winged fungus gnats (Sciaridae) were found in 98% and 96% of homes, respectively. Nearly half

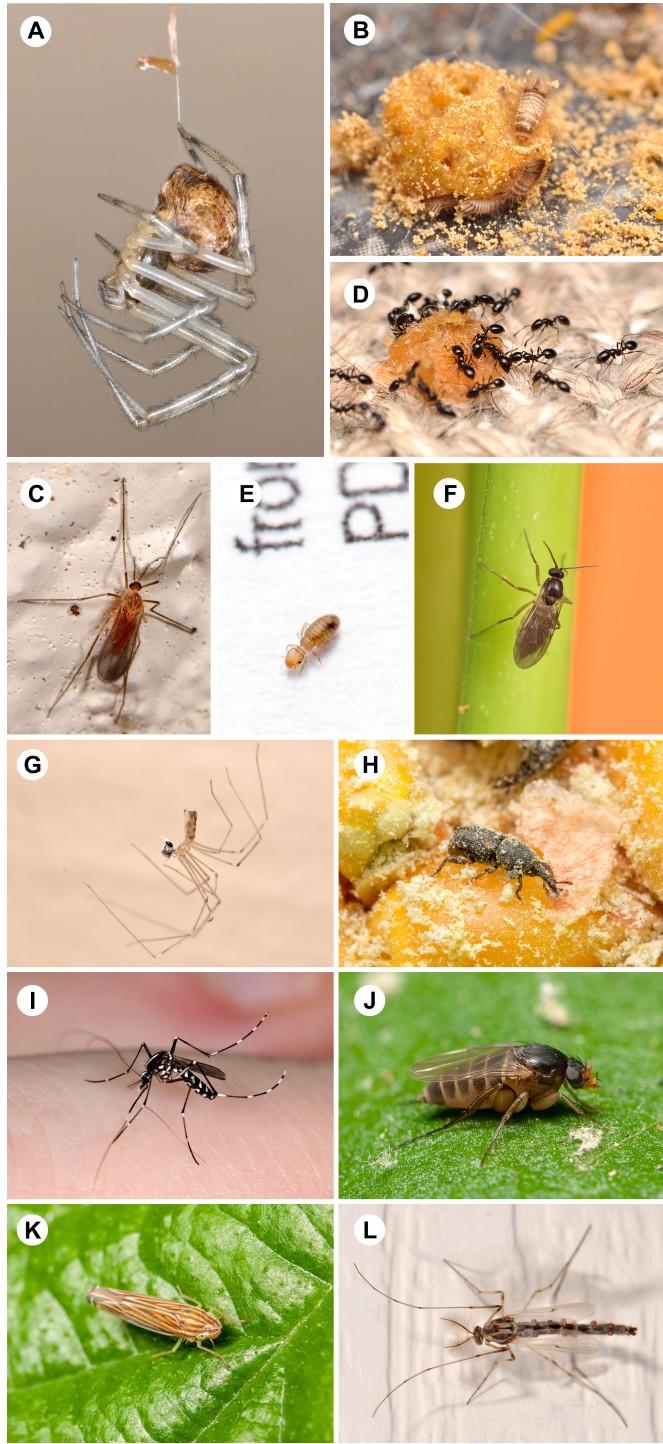

**Figure 4** **Photographic representatives of the most frequently collected arthropod families.** Twelve (12) families were represented in at least 80% of homes. For each family we present the common name and percentage of homes it was found in, followed in parentheses by the scientific family name and species level identification when possible. (A) cobweb spiders, 100% (Theridiidae; shown here *Parasteatoda tepidariorum* (Koch)); (B) carpet beetles, 100%, (Dermestidae; shown here *Anthrenus* larvae); (C) gall midges, 100% (Cecidomyiidae); (D) ants, 

**Figure 4 (… continued)**
100% (Formicidae; shown here *Monomorium minimum* (Buckley)); (E) book lice, 98% (Liposcelididae); (F) dark-winged fungus gnats, 96% (Sciaridae); (G) cellar spiders, 84% (Pholcidae; shown here *Pholcus* sp.); (H) weevils, 82% (Curculionidae; shown here *Sitophilus zeamais* (Motschulsky)); (I) mosquitoes, 82% (Culicidae; shown here *Aedes albopictus* (Skuse)); (J) scuttle flies, 82% (Phoridae; shown here *Dohrniphora incisuralis* (Loew)); (K) leafhoppers, 82% (Cicadellidae; shown here *Sibovia* sp.); (L) non-biting midges, 80% (Chironomidae). All photos by MAB.

of all families (five of 12) found in over 80% of homes were true flies (Diptera): fungus gnats (Sciaridae); mosquitoes (Culicidae); scuttle flies (Phoridae); non-biting midges (Chironomidae); and gall midges (Cecidomyiidae).

Typical household pests were found in a minority of the homes, such as German cockroaches (*Blattella germanica*: 6% of houses), subterranean termites (Rhinotermitidae: 28% of houses), and fleas (Pulicidae: 10% of houses); bed bugs (*Cimex lectularius* Linnaeus) were not found during the study. Larger cockroaches (Blattidae), such as smoky brown (*Periplaneta fuliginosa* (Serville)) and American cockroaches (*Periplaneta americana* (Linnaeus)) were found in the majority of houses (74%). However, the American cockroach (which is the only of the two considered a true pest) was only recovered from three homes; smoky brown cockroaches made up the vast majority of large cockroaches collected. All pest species were less common than other more inconspicuous arthropods such as pillbugs (Armadillidiidae, 78%) and springtails (Entomobryidae, 78%).

In addition to those listed above, many of the same pests we recovered were also found in archaeological sites (*Nielsen, Mahler & Rasmussen, 2000*; *Panagiotakopulu, 2001*; *Panagiotakopulu, 2003*; *Kislev, Hartmann & Galili, 2004*; *Panagiotakopulu, 2004*; *Kenward & Carrott, 2006*). These included grain weevils (Curculionidae: *Sitophilus* Schoenherr), carpet beetles (Dermestidae: *Anthrenus*), grain beetles (Silvanidae: *Oryzaephilus* Ganglbauer), cigarette and drugstore beetles (Anobiidae: *Lasioderma* Stephens & *Stegobium* Motschulsky, respectively), house flies (Muscidae: *Musca domestica*) and lesser house flies (Fanniidae: *Fannia* Robineau-Desvoidy).

## Arthropod distribution within the home

Arthropods were found on every level of the home and in all room types. Only 5 rooms (non-attics) had no arthropod specimens collected (four bathrooms, one bedroom). Six arthropod orders dominated houses, comprising 81% of the diversity in an average room: Diptera (true flies, 23%), Coleoptera (beetles, 19%), Araneae (spiders, 16%), Hymenoptera (predominantly ants, 15%), Psocodea (book lice, 4%), and Hemiptera (true bugs, 4%) (Fig. 5). Eight additional orders made up another 15% of the diversity (Blattodea, Collembola, Lepidoptera, Isopoda, Zygentoma, Polydesmida, Orthoptera, and Acari), while all remaining orders comprised a total of 4% of the overall diversity (Fig. 5). The percentage of rooms in which an arthropod was collected varied among taxa, as did their presence in rooms of different types (Table 1).

## Dust mite sampling

Dust mite samples contained from 0 to 421 total specimens, with an average of 38.12 (s.d. = 71.5); dust mites were found in 76% of the homes sampled (Table 1). Significantly

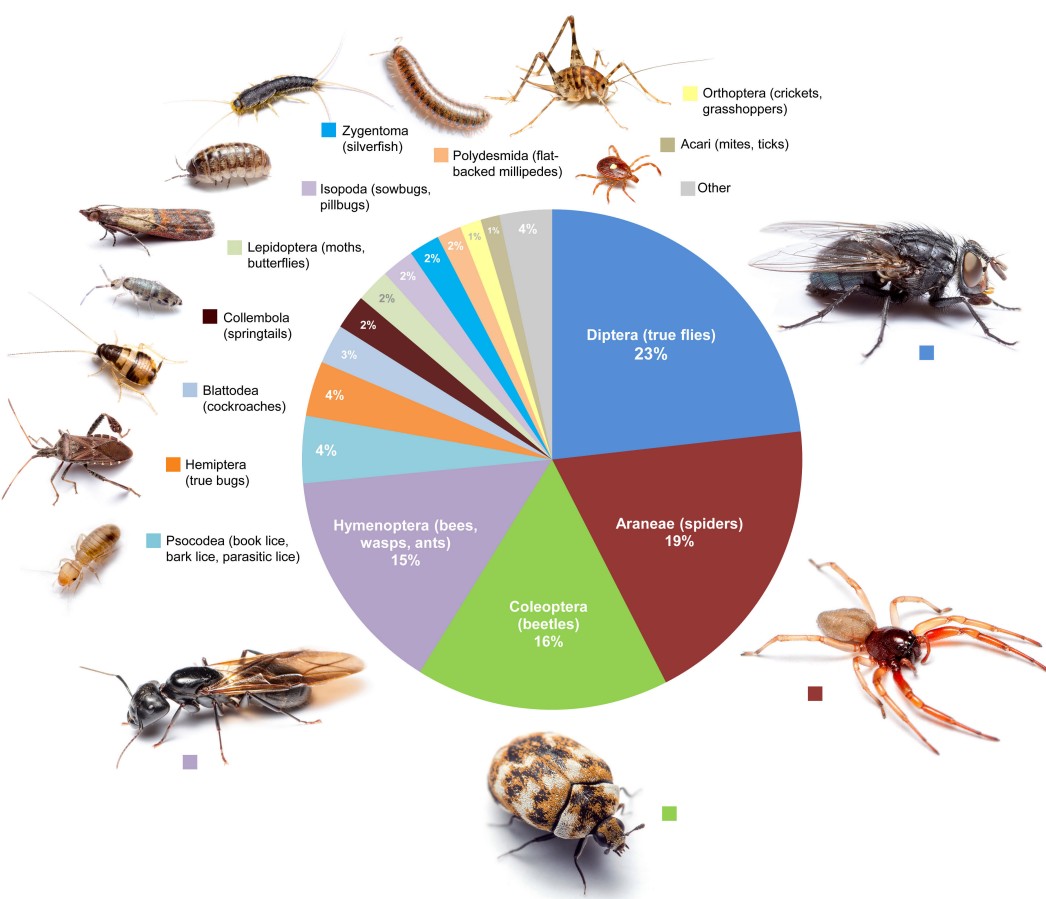

**Figure 5** **Proportional diversity of arthropod orders across all rooms.** Average morphospecies composition calculated across all room types. All photos by MAB.

more mites were collected from carpeted surfaces than hard surfaces (e.g., wood floors) (Kruskal–Wallis test: $\chi^2 = 10.692$, $p = 0.001$). Of those identified from the subset, all were *Dermatophagoides* sp. (Pyroglyphidae).

## DISCUSSION

As household pests and disease vectors, the indoor arthropods in our daily lives have had a substantial impact on human society both historically and today. Although extensive research has been done on a small number of arthropod pest species, the data presented herein represent the first comprehensive survey of the arthropod diversity collected from urban and suburban houses. In the absence of similar studies that could provide baseline data for comparison, our results are surprising both in terms of the prevalence of arthropods (virtually every room was occupied) and in terms of their diversity.

The diversity of arthropods found indoors extends far beyond commonly recognized species. We found that an individual house may have hundreds of arthropod species within it, with each house on average containing 62 families and a minimum estimate of 93 morphospecies. The true diversity among these 50 homes is undoubtedly much

higher due to limitations of the minimum estimate (it assumes no species turnover between rooms), the presence of cryptic species and our sampling method that excluded areas behind walls, under heavy furniture, and in drawers and cabinets, all of which undoubtedly serve as potential refuges for additional arthropods. While sampling 50 homes did lead to a decrease in the estimated diversity of families, clearly there are still many other families and morphospecies that are yet to be recovered and characterized from homes (Fig. 2).

We found that four groups of arthropods dominate the average room: flies (23%), beetles (19%), spiders (16%) and hymenoptera (predominantly ants, 15%) (Fig. 5). Overall, there are more types of flies associated with human homes than any other group of animals. Some flies have evolved close associations with humans, while others (Chironomidae and Cecidomyiidae) may arrive in houses as part of 'air plankton'; their presence indoors is more a reflection of their abundance outdoors than of the ecology inside homes. Despite their prevalence, flies represent only a small proportion of taxon-focused studies in the urban landscape (*McIntyre, 2000*). Recent studies have revealed new information on flies in urban landscapes, including 30 new scuttle fly species identified from urban Los Angeles (California, USA), indicating that the true diversity of these flies, and likely many other small fly groups, in human-developed areas is underestimated (*Grimaldi et al., 2015*; *Hartop, Brown & Disney, 2015*).

Book lice (Psocodea: Liposcelididae), were found to be among the most ubiquitous indoor arthropods (found in 49 of 50 houses). Book lice are close relatives of parasitic lice and have a long evolutionary history of living, among other places, in close association with birds, mammals and their nests (including those of primates; *Grimaldi & Engel, 2005*). As stored grain pests, fungus feeders, and scavengers, book lice thrive in indoor environments. *Liposcelis bostrychophila* Badonnel, for example, is a globally-distributed, anthropophilic species whose widespread success and resistance to control measures is in part due to its parthenogenesis, ability to disperse through air, wide diversity of diet, and resistance to starvation (*Diaz-Montano et al., 2014*). Book lice have become more common in houses in the United Kingdom over time (recovered from 14% of houses sampled in 1987 versus 30% in 1997; *Turner & Bishop, 1998*) and are more prevalent in areas of high humidity in houses in Spain (such as kitchens and bathrooms; *Baz & Monserrat, 1999*). However, perhaps due to North Carolina's humid climate or our sampling methods, we found book lice distributed throughout houses.

Dust mites were found in the majority of homes (76%). Previous studies have found dust mites in 30–100% of sampled homes across the US (*Arlian et al., 1992*). Human association with dust mites may have been established with the origins of dense human settlements; dust mites likely shifted from the nests of synanthropic birds or rodents to human houses (*Klimov & O'Connor, 2013*). To control dust mite populations, it is often recommended to remove carpets because they provide protection, thermoinsulation, higher humidity, and trap the food on which these mites feed (*Colloff, 1998*). As expected, we found much higher dust mite abundance on carpeted surfaces, consistent with previous research. Yet, paradoxically, the house that had the single highest abundance of dust mites within our study had a wood floor. Humidity levels and vacuuming frequency,
although unknown for this house, may explain the discrepancy. Characteristics of different wood floors, such as age and quality of build, could also affect mite abundance since gaps between boards can provide habitat for mite populations.

Because previous studies of indoor arthropods have largely focused on pest groups of economic and human health importance (e.g., *Runstrom & Bennett, 1990*; *Colloff, 1998*; *How & Lee, 2010*; *Crissman et al., 2010*), we expected common pests to be among the most frequently found groups of arthropods in the homes. In fact, we found a relative dearth of typical household pests. The only exception to this was the prevalence of the smoky brown cockroach, a species that is not truly considered pestiferous because it does not generally develop pest-level populations in homes due to its need for high humidity and moisture (*Robinson, 2005*). It may be that we collected more specimens of this species due to their intolerance and ultimate death within the homes. While the lack of many pest species could be an artifact of the sampling design (sampling for species occurrence rather than abundance, as well as sampling in free-standing homes rather than other forms of human habitation such as apartments, townhouses, etc.), it appears that the vast majority of arthropods that live among us cause no direct harm. Unfortunately, many insects and arthropods we collected are considered pests based solely on their presence in the home (i.e., nuisance invaders; *Hahn & Ascerno, 1991*; *Cranshaw, 2011*), despite having no direct impact on people or their possessions.

Many arthropods we identified from houses were unexpected—either in terms of the frequency with which they were found or because they are rarely found outdoors, much less indoors. Gall midges (Cecidomyiidae), although found in every house sampled, were not even mentioned among the over 2,000 species listed in a recent compilation of urban insects and arachnids (*Robinson, 2005*). Leafhoppers (Cicadellidae), as plant feeders, are not associated with the indoor biome (*Robinson, 2005*), yet were among the groups most frequently found in houses. Moths and butterflies (Lepidoptera), on the other hand, were collected infrequently, making up only 2% of the average diversity in a room; this is disproportionate to their known overall diversity (the order comprising over 10% of described insect species; *Heppner, 2008*). Although ants (Formicidae) were expected and found in 100% of houses, further identification at the genus and species level revealed taxa that are not typically thought to occur in homes. Camel crickets (Rhaphidophoridae) are known basement dwellers in the Southeastern US, but our sampling confirmed previous reports that an invasive species, *Diestrammena asynamora* (Adelung) predominates over native species (*Ceuthophilus* spp.) (*Epps et al., 2014*). Other unexpected finds were ant-loving crickets (Myrmecophilidae), the smallest orthopterans (*Whitman, 2008*), which were found in homes with ant infestations; beetles from the relatively rare suborder Archostemata (families Cupedidae and Micromalthidae); and a larval beaded lacewing (Berothidae), a rarely seen neuropteran known to live within termite nests where they paralyze termites with an airborne chemical before feeding on them (*Johnson & Hagen, 1981*).

Of the arthropods we found that live out a portion of their life cycle in human houses, there is a broad diversity of trophic levels and life histories represented. Apart from a few herbivorous arthropods associated with houseplants or those inadvertently living

indoors (for example, brought in on cut plants), most taxa sampled from houses were either scavengers, predators, or parasitoids. Carpet beetles (Dermestidae) were found feeding on dog kibble, dead insects and nail clippings. Other scavengers, like silverfish (found in 68% of homes) and book lice were also common. Carrion-feeding flesh flies (Sarcophagidae) were found during the study emerging from a rodent killed by a house cat. Spiders (including spitting spiders, Scytodidae, that spit venom up to a centimeter to ensnare prey; *Foelix, 2011*) and centipedes (especially Scutigeridae) were the primary predators sampled. Minute parasitoid wasps (especially Eulophidae and Platygastridae *s.l.*) that potentially parasitize other household arthropods were also common inhabitants. For instance, one species of Eulophidae, *Aprostocetus* (*Tetrastichodes*) *hagenowii* (Ratzeburg) (Table S1), a known parasitoid of blattid cockroach egg cases (oothecae), was commonly collected in homes as were its hosts. Considering the range and abundance of life histories found in our study, the trophic dynamics of the indoor ecosystem is an area in need of future study.

The rich arthropod diversity we identified from houses reflects a gradient of association with human habitations, from synanthropic arthropods that appear strongly adapted to human houses (cobweb spiders, carpet beetles, book lice), to others that seek shelter and resources only on occasion (ants, ground beetles, hunting spiders, smoky brown cockroaches), to many groups that simply become trapped in houses to their own detriment (leafhoppers, gall midges, click beetles). Most of the arthropod groups we identified do not have life histories that are known to be closely associated with the indoors. Many arthropods may find themselves indoors as a result of the 'Malaise trap effect': houses, like Malaise and other flight intercept traps, are effective at capturing local arthropods that may be travelling through the environment or are attracted to houses by artificial light, food, and shelter. These arthropods may be active in a house for a short period of time, where interactions between them and the house's residents may occur, but eventually they must either find an exit or succumb to mortality. The idea that homes are traps or filters of local, outdoor arthropod fauna implies the importance of further investigating the dynamics between the greater landscape and the indoor environment.

Biodiversity in urban landscapes is richer than was once thought (*McKinney, 2008*; *Fattorini, 2011*; *Fattorini, 2014*), and we find here that the indoor, manufactured environment also supports more diversity than anticipated. These findings represent a new understanding of the makeup of the indoor arthropod community and their distribution within houses. Arthropods within our homes are both diverse and prevalent, and are a mix of closely synanthropic species and a great diversity of species that wander indoors by accident. Many species we found were unexpected, unnoticed by residents until they were collected, and play no pestiferous role in human houses. Yet, further research on the ecological dynamics of the indoor biome is needed to understand the potential economic and health implications of the species that live and have evolved in such close proximity to us.

## ACKNOWLEDGEMENTS

We are very much indebted to the many volunteers who allowed us to sample the arthropods in their homes. Kelly Oten aided in designing many aspects of the study and participated in sampling homes. We also thank others who helped collect specimens: Nancy Brill, Mary Jane Epps, Clint Penick, Amy Savage, Patricia Turner, and Steven Turner. Melissa Howell aided in sorting arthropods.

### Funding

This work was supported by National Science Foundation funding DEB 1257960 and NSF Career 0953350. The funders had no role in study design, data collection and analysis, decision to publish, or preparation of the manuscript.

### Grant Disclosures

The following grant information was disclosed by the authors:
National Science Foundation: DEB 1257960.
NSF Career: 0953350.

### Competing Interests

The authors declare there are no competing interests.

### Author Contributions

- Matthew A. Bertone conceived and designed the experiments, performed the experiments, analyzed the data, wrote the paper, prepared figures and/or tables, reviewed drafts of the paper.
- Misha Leong analyzed the data, wrote the paper, prepared figures and/or tables, reviewed drafts of the paper.
- Keith M. Bayless reviewed drafts of the paper, specimen Identification.
- Tara L.F. Malow analyzed the data, data Entry & Management.
- Robert R. Dunn conceived and designed the experiments, contributed reagents/materials/analysis tools, reviewed drafts of the paper.
- Michelle D. Trautwein conceived and designed the experiments, analyzed the data, contributed reagents/materials/analysis tools, wrote the paper, reviewed drafts of the paper.

### Data Availability

   Data has been uploaded to the Supplemental Information.

### Supplemental Information

Supplemental information for this article can be found online at http://dx.doi.org/10.7717/peerj.1582#supplemental-information.

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
