# Peer review of "Arthropods of the great indoors: characterizing diversity inside urban and suburban homes"

_PeerJ, doi:10.7717/peerj.1582_

## Round 0.1 · original submission · Major Revisions

Dear Matthew,

As you will notice, I received very different assessments of your work. I still think it is a valuable and interesting study, but there are several important issues that have to be addressed before it is ready for publication, particularly regarding the used analyses.

·

Basic reporting

No comments

Experimental design

No comments

Validity of the findings

No comments

Additional comments

This is an interesting and well executed study on a so far overlooked, but important subject in urban ecology. I think the paper will represent a good contribution to the advancement of our understanding of human-insect interactions in urban areas.
Despite the obvious importance of indoor arthropods, their diversity has been so far largely unexplored, most of available researches dealing only with pest species. The impressive arthropod diversity recorded in this study indicates that, in contrast with the idea that only few arthropod species live in our houses, the indoor biome (to use the suggestive expression recently popularized by Martin et al., 2015) hosts a large number of non-pest species.
I found the ms very well organized, clear and complete. The Introduction concisely presents a good summary of the state of the art in the study of arthropods associated with houses and clearly presents the aims of the study. Methods are simple, but appropriate. Results are extensively and clearly presented. Discussion place authors' results in the framework of current literature.
I have only few suggestions to improve ms readability and impact.
Line 69-73: I suggest to expand this section with reference to a recent paper by Martin et al. (2015): Martin, L J. et al. (2015) Evolution of the indoor biome. Trends in Ecology & Evolution, 30: 223 – 232.
Line 75: Add a reference to support this assertion about Psychodidae. Or say "are probably less"
Line 79: ...and, in some cases,…
Line 86 …( and even unnoticed) … [delete comma]
Line 87-88: … be it beneficial, neutral or negative…
Line 143: It is not clear what "identified past mite” really means
Line 157: Is this the assumed minimum estimate? Please, clarify.
Line 174: Please, add version, author, etc of EstimateS software and give it in the references section
Line 182: I see that you only report the coefficient of correlation, not the regression equation. If you are interested only in the straightness of correlation, and not in regression parameters, you should say that you used “correlation”, not “regression” (although the r-statistics are obviously the same). Are the data normally distributed? And are the relationships linear? If you are only interested in correlation, please consider the possibility of using a Spearman rank correlation, instead of Pearson r. Spearman rank correlation which does not have assumptions about normality and linearity.
Line 219 ff: replace x with the Greek symbol of Chi
Line 225: Rephrase as follows: “The number of arthropod families found in a room increased with the number of windows…..”. F-statistics are unnecessary, only report r and p.
Line 227: Rephrase as follows: “….families increased with the number of doors…”
Line 401: Malaise (use capital letter; Malaise is the name of inventor of this trap, René Mailase)
Line 422: Sorry for shamless self promotion, but - in addition to McKinney (2008) - I suggest to cite two more recent papers:
Fattorini S. (2014) Urban biodiversity hotspots are not related to the structure of green spaces: a case study of tenebrionid beetles from Rome, Italy. Urban Ecosystems, 17:1033–1045
Fattorini S. (2011) Insect extinction by urbanization: a long term study in Rome. Biological Conservation, 144: 370–375.

Reviewer 2 ·

Basic reporting

There are a couple of issues in this area, foremost in regard to the literature review of this field as presented in the introduction as well as missing expectations/hypothesis. Also parts of the discussion should be moved to the results section.
See general comments for details.

Experimental design

this is good but the catgorization of rooms needs clarification

Validity of the findings

The findings are robust although I do have some doubts regarding the calculations of minimum/maximum morphospecies numbers. Again, some references would do good, to establish the validity of analysis method.
Overall the conclusions need to be linked more clearly to the previously stated expectations/hypotheses.

Additional comments

I generally like this inventory of indoor biodiversity, it is a newly emerging field of study which would benefit from such baseline inventories. Nevertheless, I think this manuscript needs a more thorough revision, especially in regard to the questions under investigation. As it is now, it is a tough read, with many lengthy sections that solely describe results and lack references to other studies as well as own expectations. I think this manuscript would greatly benefit if a couple of things are clarified:
(1) What are the predictions that this study is based on and the results could be evaluated by. I am aware that only few publications are available in the field. Nonetheless, I believe that from the ecology of the species or families under investigation some expectations are possible and should be clearly stated in the introduction.
(2) The issue of economic and human health impact are mentioned in the introduction. I would generally appreciate a stronger focus on this issue. The last paragraph of the discussion picks up this issue again, but in between it is rarely mentioned. Either elaborate more in this subject throughout or put less focus on it in the introduction.
(3) I think, this manuscript needs other lines of argumentation, to put together the loose pieces that are encountered while reading. This could be in line with the expectations above. Further down I also bring into question the categorization of rooms. I asked myself what this is based on. I listed a couple of examples, which came to my mind. Maybe, if you use these, for a start, you could build your expectations around them and use them again in the discussion to structure you manuscript. Maybe this could help to answer if the most diverse room is the one with highest human traffic, the most predatory species live in the kitchen etc…
(4) Move as many results, currently in the discussion section, into the results section as possible.
(5) What about literature in the field of urban arthropods, or even pest management and so on? Generally, there were very few references, especially in this regard.


Title: in my eyes, the ‚all taxa‘ approach is misleading and leads to false expectations of the reader. The arthropod inventory is sufficiently interesting.
Abstract: pest species are mentioned frequently, as well as their impact, economic as well as health, on humans – this argumentation should be further developed throughout the paper, it is largely neglected. I am missing comparisons between pest species and non-pest species etc.
L 52: this should receive citations, especially when referring to the indoor biome.
L 60-61: Humans have surely been infested with parasites ever since they emerged, and their ancestors before have to. It would be interesting to give a little more information on how the spectrum of arthropods, that humans have been exposed to since they first began to settle, varies in comparison to other great apes.
L 67-68: also here, I think some examples of what these are would be good.
133: Please give reason why attics and crawl spaces where sampled in different intensity
L 148-152 Can you provide numbers here? How many did you identify to species level, how many samples were larvae, how many female Sciaridae, etc. …
L 153: You say here, that you could not compare morphospecies between rooms. Mention here, that you are referring to their taxonomic identity, as later on you are comparing morphospecies numbers between rooms.
L 146-162: can you provide citations for this methodology and from this derive a hypothesis, which one of the two estimates the true number is closer to, and what factors this might depend on (abiotic, biotic…)?
L 165-172: What did you base this categorization on? The number of people these rooms are used by, the resources to be expected here, the predominant microclimate, etc..? Specifically, I do not understand the grouping of bedrooms together with offices and libraries. This needs clarification. Does this categorization allow you to come up with hypothesis regarding a rooms’ category diversity, or which families should be present only, or mostly, in any one type of room?
L 183 mentions dust mite abundance. Were the abundances of other arthropods also recorded? Or was this a special case for dust mites due to sampling method. If so, please briefly mention this here…
Methods: Generally, I am missing citations of other studies that have applied a similar study design or analysis.
L 193: What is this number actually referring to? Is it the conservative estimate of morphospecies as mentioned in the methods? I am not quite sure, please clarify. Also, how does this relate to the number given in L 197?
Fig. 4 This invites to use a correlation of size and estimates
L 217 – 218: This should be mentioned early on in the results, or maybe in the methods section.
L 237: Maybe elaborate on the effects they have had in more detail, also, I am missing citations again.
L 242-243: I do not quite understand where the surprise comes from? If there is no baseline data to compare it with, a deviation from the expected is not possible…Please clarify this here. Maybe your expectations should be formulated more clearly in the Introduction.
L255-266: This would be a good place to come back to economic and health impacts, as mentioned in the introduction. Do the most dominant taxa pose a threat to humans?
L270-271: start this paragraph with the 23% and then go into details…
L285-286: please clarify this assumption
L 365: This should clearly be mentioned earlier in the introduction
L 369-372: Please clarify here already, that American cockroaches were only found in three homes
L 382-389: This is a very interesting finding! I think this manuscript would greatly benefit from a more detailed description here together with mentioning potential reasons – have human settlements changed so little since then from an arthropods point of view?
L 398-399: This statement surprises me, as in the previous paragraph there seem to be many which have been associated for so long. Please clarify what this statement is based on specifically.
L 407 is this your hypothesis? If so, I would specify it, otherwise give a citation. Also make sure to mention this hypothesis in the introduction
L 410: this prediction, along with the others first mentioned in the discussion should be more clearly stated in the introduction
L 424 Please provide reasons for this anticipation

·

Basic reporting

• The article should include sufficient introduction and background to demonstrate how the work fits into the broader field of knowledge. Relevant prior literature should be appropriately referenced.

Generally agree, but with the following comment:

I would like to have seen reference to some of the wider urban biodiversity literature. Whilst the reference to archaeological literature was interesting, I was not sure exactly where this fitted in - perhaps if the research question(s) had been explicitly stated this would have been more obvious. The stated aims are essentially descriptive. I think a clearly worded hypothesis would help clarify the relation of the manuscript to the wider urban ecology/entomology field.


• Figures should be relevant to the content of the article, of sufficient resolution, and appropriately described and labelled.


Generally disagree, with the following comments:

Table 1: Under the heading "Acari" - there is label reading "UnID Oribatida". Change to "Oribatida" (or explain why this has UnID and nothing else). There are some formatting changes in the Coleoptera section (font size). I suggest removing the "common name" column as this doesn't add much and these 'common names' do not translate well geographically. Given the intended audience, I don't think common names are required.

Figure 2: I was missing a legend for this figure. X-axis label should be given in full (e.g. no. of houses sampled).

Figure 3: Remove the "maximum number of morphospecies" dots and just use the estimated species richness values. Please see "Analysis" comments for rationale.

Figure 4: Whilst this is a nice figure (they are good photos), I don't think it adds much to the analysis and question whether this needs to be included.

Figure 5: Again a pretty figure, but better suited to outreach materials. This would perhaps be more appropriate if re-drawn as a Rank Abundance Distribution (aka "Whittaker plot"). Rank from most to least abundant along X-axis, Y-axis is log scale of proportions relative to most common species. See Magurran (2004) for an example of this plot.

Figure 6: X and Y axis label missing.

Figure 7: I suggest scaling the figures in R so that the font size of labels is consistent. Whilst I am familiar with how R draws these and the notation for outliers (*), it would be worth mentioning that this is what these are in the legend. Were these outliers manually identified or using the default R detection?

Figure 8: I may have missed this, but I felt this figure seemed to come out of nowhere – I didn't see mention of this, or the details of the transformations to meet normality assumptions, in the methods text. See my general comments on the analysis below.

Figure 9: X-axis label missing. A personal preference here, but I think the use of coloured boxplots is a no-no. I'm not sure this particular result is in need of a figure in the main text – quoting the test statistic and p-value should be enough as the difference between the two 'treatments' is very large and distinct.



• All appropriate raw data has been made available in accordance with our Data Sharing policy.

Generally agree, but with the following comment:

I suggest that you format the species list as a separate species x site incidence matrix. This is the most common format for biodiversity analysis and would facilitate future meta-analysis.

Experimental design

• "The submission should clearly define the research question, which must be relevant and meaningful."

Disagree.

This was missing as far as I can see.


• "Methods should be described with sufficient information to be reproducible by another investigator."

Disagree, with the following comments:

A number of small omissions were made here.For the work to be truly repeatable, the same houses would need to be re-sampled, so perhaps a supplementary table of the GPS locations of each property centroid would be useful.

Whilst the arthropod specimens were not collected according to an effort-standardised protocol, it would be useful to understand how many specimens were collected in total from each house, and from each treatment (room type).

Line 109 "local variables of interest" – these are never stated (?).
Line 111: Please report the sampling dates in full. Does this also mean there were repeated sampling visits to each house, or were some houses sampled in May, and then others through the season to October? This could be a significant confounding variable.
Line 115: There is no mention of sampling effort here. Did you assume sampling to exhaustion, or was each room timed? How have you assessed that sampled was sufficient for further analysis in each treatment?
Line 138: This may seem obtuse here, but in my house the bed is in the middle of the room. Did you move furniture to sample if necessary? If so, might this have some effect compared to rooms where the middle of the room was not covered with furniture and perhaps hovered or walked on more often? Was change in 'cover' recorded?
Line 142: How much is a subset?
Line 173: I assume sample-based methods were used, but this should be stated explicitly.

Validity of the findings

• The data should be robust, statistically sound, and controlled.
• The conclusions should be appropriately stated, should be connected to the original question investigated, and should be limited to those supported by the results.

Disagree, with the following comments:

My understanding of the analysis presented would suggest this is not the case. I do not believe the authors have fully accounted for differences in sampling effort, and fully justified how rooms in the same house can be taken as independent treatments.

For example (line 174): "For each room we calculated the total number of morphospecies and families collected … and tested for patterns …" As each room was sampled using different amounts of effort, comparisons of observed taxon richness could reasonably be expected to simply reflect differences in effort and detectability. None of the current analysis should use the observed data, which should be presented only for completeness as supplementary material (and retained as voucher specimens).

In lines 204-206 it is stated: "Considering that our conservative species estimate assumes that rooms with the greatest number of morphospecies by family included all species from other rooms, this number is likely much lower than the true number of species per house". If I understand this correctly, this is similar to "average richness per taxon scores" approaches which have been shown to be prone to various theoretical pitfalls (Gotelli and Colwell, 2011).

I have tried to interpret the analysis as addressing various possible questions and in these cases found the analytical approach taken by the authors to be inappropriate. It is possible that I simply have not identified the intended research question correctly, but I suspect the analysis is still limited by the use of multiple pairwise comparison rather than a single multivariate analysis (such as GLMM).

My recommendation is that the authors clearly define the question being answered, define what is being compared (e.g. estimated number of families in each treatment - not "diversity" or "species richness" which mean different things to how I think they are being used here) and make the distinction between estimated and observed taxon richness data very clear.

If I have identified the question intended (e.g. something like "are there differences in arthropod family-richness between different rooms in houses?") then the analysis should be re-done in the following manner and conclusions verified:

1. Construct sample-based rarefaction curves for each room category (n=50 rooms per category). Plot on same figure and look for overlap of 95% confidence limits. You could look at using the ICE estimator, which places more emphasis on rare "species" (in <10 samples) and might be a useful comparison against Chao2 which is more concerned with estimating the minimum number of 'species' in the sample set.

If there is no overlap between room categories, then multivariate approaches should be used to investigate the effect of room type with regard to all the other measured variables (e.g. S(est) ~ house area + room size + carpet (0/1) + floor type + interactions, etc., with something like "house ID" or "date sampled" as a random effect and using suitably transformed variables).

The authors would still need to state justification for treating rooms within a house as independent statistical units, as if there is 'assemblage share' between them then the analysis would be somewhat meaningless. Perhaps one way of doing this using their data would be to quantify the number of shared species (using Jaccard's or similar) between rooms for only the taxa where you identified these to species level in all houses. If the % shared species is low, then I would be more inclined to believe that the rooms could be treated as independent. Even this is problematic however, as the components of beta-diversity are linked to alpha diversity, which is in turn influenced by sampling effort.

Figure 8: Linear regression (number of windows ~ S): There is another issue here with sampling in that you are comparing different sample sizes and areas = effort. Presumably a house with 30 windows was sampled 30 times more than a house with 1 window, if you looked at all the windows. So it is no surprise the taxon richness increases with number of windows, since you looked harder, and the area of "window habitat" for things like spiders living on the windows increased 30 fold. This cannot be a straight line relationship either - this is basically a plot of sampling effort (number of windows) against taxon richness, so will eventually plateau the same as any other species accumulation plot.

Linear regression using untransformed observed data is therefore not appropriate for this analysis. See Gotelli and Colwell (2011) (http://onlinelibrary.wiley.com/doi/10.1046/j.1461-0248.2001.00230.x/pdf) for further info.

Additional comments

The authors (Bertone et al.) present an interesting and taxonomically comprehensive survey of the arthropod biodiversity associated inside domestic houses. I was impressed by the scope of the taxonomic work, which included some quite tricky groups (even to family level). The manuscript is generally well written and accessible, although improvements to the clarity could be made in key areas, and an explicit statement of the hypothesis tested would help significantly.

I think some of the discussion points were quite relevant to wider urban conservation issues and presented interesting points for urban planning and conservation (e.g. the perception that most arthropods in houses are pest species, when it seems they are not; would wider knowledge of this by the public help promote or hinder urban arthropod conservation?).

My major concerns however lie with the analysis of the research, of which the aims are unclear and the methods appear inappropriate / violate assumptions of the the methods. In particular, the use of observed richness measures (without controlling for effort), the linear analysis of non-linear data and the lack of multivariate approaches make me question the robustness of the conclusions.

In summary, I feel I must suggest that this manuscript should be rejected due to fundamental flaws in the analysis. I strongly suspect that with appropriate re-analysis, the reported findings would not be supported, and as such the theme of the paper would be significantly different. In this case, submission of a new manuscript with suitable focus would be more appropriate.

Omission: I did not see reference to voucher specimens, or where these were deposited. Please see the recent paper by Turney et al. 2015 for rationale (https://peerj.com/articles/1168/).


Specific points:

Line 52: "… is represented by Arthropods" should read "are Arthropods."
Lines 75-80: There seems to be some conflict in tense here and I think this could be re-worded to improve this.
Line 98: "general distribution" – I suggest "spatial distribution" or another suitably specific term, as when I first read this I wasn't sure what you were referring to.
Line 110: "whilst holding large scale differences in species pools relatively constant" – perhaps this is just me, but I found this difficult to understand exactly what was meant here. I assume the argument is being made that houses close enough to town centre such that variables generally measured at larger spatial scales (such as urbanisation density and housing age) were not too dissimilar? I think this could be re-worded for clarity.
Line 143: "past mite" should be reworded to "beyond Ordinal level" or similar.
Line 146: A small point, but if you re-word "… to the taxonomic level of family" to "We identified all specimens to family level" it reads better and is more succinct.
Line 149-151: Perhaps I am being overly pedantic, but I don't like the use of "wrong" here to describe identification issues relating to sex and life-stage. I think you could encapsulate all these issues by simply stating that material was identified to as fine a resolution as possible depending on what keys you had available. Most entomologists will know that you can't ID larval forms easily, or certain sexes, etc. I don't think any reasonable person familiar with sorting arthropod samples would criticise your effort in sorting this much material.
Line 155-156: You should make reference to the Oliver and Beattie paper (http://onlinelibrary.wiley.com/doi/10.1046/j.1523-1739.1996.10010099.x/abstract) where they propose this method, and perhaps a better term would be "Recognisable Taxonomic Unit" (RTU's) since you did not get all specimens to "species" level.
Line 165: how did you decide some of these definitions? I think geography is important here – I'd not come across "mudroom" before as in the UK we'd just use the porch, conservatory or hallway.
Line 187: There seems to be a formatting issue here (a tab space).

---

## Round 0.2 · Minor Revisions

Dear Matthew,

I just received the reviews. I believe that the manuscript has indeed improved since it was first submitted, and I'm happy with the thoughtful revisions by all of the reviewers.

Two of the reviewers are pleased with your revision, but reviewer 3 still has some important issues that need to be addressed.

·

Basic reporting

The paper is basically OK

Experimental design

The experimental design is OK

Validity of the findings

Findings are OK

Additional comments

I found this new version of the paper greatly improved. The authors have successfully introduced all my suggestions/corrections.
In general, I agree with the authors that this is an exploratory and hence descriptive research, not aimed at testing a priori established hypotheses. As regards my previous concerns about regression analyses, I appreciate that the authors – in response to a critique moved by another reviewer – have now analysed the data using generalized linear mixed models, which are an interesting addition.
Finally, I have only a few absolutely minor corrections/suggestions.
Line 185: Please, add a full stop (and a space) after “test”.
Line 255: I think that this x2 is, in fact, a chi-square. Please, use the correct symbol. Also, I think it would be useful to recall here that this is the result of the Kruskal-Wallis test. So, I suggest to say: “(Kruskal-Wallis test: χ2 = 10.692, p= 0.001)”.
Table 2: I think that 3 decimal places can be enough.
Thank you for this intriguing research.
Simone Fattorini

Reviewer 2 ·

Basic reporting

This article has greatly improved and is now well structured and easy to read.

Experimental design

Research questions have been clarified and aim to investigate a totally new field of urban biodiversity.

Validity of the findings

Conclusions are in line with research question and results.

Additional comments

Great revision!

·

Basic reporting

The manuscript is generally improved from the earlier submission, but I have a number of comments to raise.

Do you state a hypothesis anywhere? I assume this was something about most arthropods not being pest species or occurring in different rooms, but I don't see any mention in the introduction. I think I missed this first time around as well.

I noticed a small number of minor issues/omissions:
Line 167: It states that you had 6 categories (I actually count 7, including the "other" category) and that "other" rooms were excluded from the analysis - yet in table 2 it lists a category as "Room type - other".
Line 170: It was a bit unclear here. So the number of morphotaxa per house (sample) - was this from the minimum number for the house (total overlap between rooms) or the maximum (zero overlap between rooms)? I think in the discussion it seems to mention the minimum, but worth clarifying this in the methods first.
Line 185: A typo/formatting error; "Kruskal-Wallis testAll analyses".
Line 276: "We find that" – should this not be "We found that"?
Line 320: "… sampling for species diversity" – perhaps more accurate would be "species occurrence". Your sampling was geared towards incidence data (reported as numbers of morphotaxa per house), and you don't really quantify biodiversity in the traditional sense (e.g. alpha/beta indices, evenness, etc.).

Figure 2 seems to lack a figure legend.

Figure 3. This is surplus to requirements now you have the accumulation curve in figure 2. In general, the rarefied/extrapolation approach is more robust, and your confidence limits essentially encapsulate the min/max thing you did before in figure 3. Unless, perhaps, you wanted to ask a question along the lines of "how does house size influence the num. of taxa collected?" (and linking to established theory in species-area relationships perhaps?). The question or relevance of house size does not seem to be mentioned anywhere else in the manuscript, although I may have missed this (the fact this rises with bigger houses, and you presumably sampled bigger houses more, means this is really a "species-area-sampling-curve").

Figure 5: I would like to revisit this (following comments in rebuttal) – A rank abundance plot conveys more for less – not only the relative proportions of each family, but the rank order of species commonness. It also shows at a glance the evenness of your assemblage/community – and the shape of this "curve" is often informative. Lastly, it also tells the viewer how many taxa/species you collected.
I would consider under "Figures should be relevant to the content of the article", and given that this manuscript is relating to biodiversity, that a standard biodiversity plotting method should be adopted. It is superior to a pie chart in many ways.

Experimental design

"Methods should be described with sufficient information to be reproducible by another investigator"

I make some specific comments/highlight omissions under the validity of the findings section and basic reporting, but otherwise have no additional comments for this section.

Validity of the findings

Unfortunately, I do have some more concerns here.

I think the authors, as they state in the rebuttal, are aiming for a descriptive manuscript. The initial description of the species/family occurrence (lines 190-243) poses no real concern. Likewise, much of the discussion relates to this material and again I think this is fine. The finding that houses have a generally high number of RTU's in them, and that most are non-pest species, is interesting. That some families were common and others rarer is also interesting. The result in figure 2 (accumulation curve) that the discovery curve is flattening is also interesting as it gives some idea about the potential richness of the system and will help reduce any concern over the level of sampling (which was substantial - but the levelling off is a good result). I think a discussion of these general observations would make a good self-contained research manuscript and would be readily publishable.

My concern arises from the analysis of mechanisms (or predictors) relating to explaining the patterns of biodiversity (lines 244-250). I also have concerns over the lack of clear identification (or investigation) of auto-correlated variables and similar statistical issues. I feel that by trying to tackle this sort of (surprisingly complex) question, the authors are opening themselves up to criticism that perhaps the original goal of the research was never designed to address.

I provided some advice in my previous review of the manuscript, and it seems some of this has been attempted, but I am not clear on exactly what has been done where - e.g. where observed vs estimated S has been used, how models were fitted or simplified, how independence of room type was justified (e.g. accumulation curves for each room category or floor, then non-overlap of 95%CL), etc.

I suggested making sample accumulation curves (for each category of room) as these ensure fair comparison at a standard level of effort (or at least as fair as is possible). If there were differences in accumulation curves between rooms, (or floors), then this would be a robust result and would facilitate later GLMM analysis to determine predictors, etc. You don't actually need p-values for this sort of comparison, you just plot the curves on the sample figure and see what doesn't overlap using the 95% CL. It seems odd you would not report the accumulation curves if you had done them - and if you did not do them, then why not?

Regarding the GLMM, I was also a bit confused by the use of the "number of morphospecies collected in each room" as a response variable against "room type" and "floor level". Did you mean the number of RTU's in the house? I tried to write your model out from what was given in the methods section and came up with something like this:

S(obs. in each room) ~ room type + floor level + floor type + (continuous variables) + 1~|house

Is this about right? This would mean part of your response variable (S obs. in each room) is nested within your predictor (room type) - "each room is a type of room". You'd also perhaps expect 'room type' to show an interaction with the floor of the house (bedrooms are typically upstairs, etc.), but I did not see mention of testing for interactions.

So I think something is wrong with the way the GLMM has been set up that has resulted in a model where almost every variable is very significant and has very small error. This doesn't feel right - having run this sort of analysis in the past, it is often the case that 12-20 measured variables will be reduced to 2-3 that are actually significant predictors of biodiversity. You have variation in your data, so this should be apparent in the model analysis. Without any info in the text about the model used, any simplification performed and how they were compared (e.g. AIC scores) it's hard to tell exactly what happened however.

You also make the point in line 205/206 that the "true number of species per house" being lower when comparing figure 2 and 3. It probably is an under-estimate, but you can't really compare between these two figures as they make different assumptions -figure 2 is a re-sampling curve whilst figure 3 is a sort of "range estimate". My advice would be to ignore figure 3, and use figure 2 which uses a much more robust analysis approach. You can state how many families you actually processed in the results text (based on the specimens), but you can't really start to "guess" how many families there might be assuming no overlap. This is not biologically sensible - and what is the cut-off of error? Does it matter if you are 10, 20 or 30% off? There's no way to really know, so what does an upper estimate add if it is devoid of any confidence?

Additional comments

I appreciate the effort that has been put in to the identification and think that it presents an interesting angle hitherto understudied in the urban ecology literature. I can only hope my comments will be interpreted as supportive (albeit identifying serious issues) rather than dismissive - I think that with some modification this will be a useful addition to the literature.

Unfortunately, the analysis presents several new (or yet unaddressed) issues that represent serious issues and would prevent my recommendation of acceptance for publication without revision.

I feel that as the authors are trying to publish a descriptive paper aimed at a general audience, it would be beneficial to keep things simple - the summary of what organisms were generally common in the 50 houses, how many families you encountered (vs. what was predicted from the accumulation curve) and the notion that many were non-pests - these are interesting findings and would represent a nice self-contained descriptive study.

I do not feel the analysis of pattern - e.g. predicting biodiversity in houses in relation to the room type or carpets etc. - adds much, except opening the manuscript to criticism that I suspect the original goal of the research was never intended to answer. As I hope I have conveyed, the analysis of biodiversity in this manner is non-trivial and mistakes can easily creep in - I have only spotted many of these because I too made similar mistakes when beginning my study into this area.

I took a considerable time over my decision, and feel that major revision is appropriate. My rationale is that unless the material attempting analysis of pattern is omitted from a revised manuscript then further focus on the analysis will be necessary, which would warrant a further look.

---

## Round 0.3 · accepted · Accept

Dear Matthew,

Despite the long "gestation", I'm happy with the final result.